# Cardiac muscle thin filament structures reveal calcium regulatory mechanism

Yurika Yamada [1], Keiichi Namba [1,2,3]* & Takashi Fujii [1,3]*

Contraction of striated muscles is driven by cyclic interactions of myosin head projecting from the thick filament with actin filament and is regulated by $Ca^{2+}$ released from sarcoplasmic reticulum. Muscle thin filament consists of actin, tropomyosin and troponin, and $Ca^{2+}$ binding to troponin triggers conformational changes of troponin and tropomyosin to allow actin-myosin interactions. However, the structural changes involved in this regulatory mechanism remain unknown. Here we report the structures of human cardiac muscle thin filament in the absence and presence of $Ca^{2+}$ by electron cryomicroscopy. Molecular models in the two states built based on available crystal structures reveal the structures of a C-terminal region of troponin I and an N-terminal region of troponin T in complex with the head-to-tail junction of tropomyosin together with the troponin core on actin filament. Structural changes of the thin filament upon $Ca^{2+}$ binding now reveal the mechanism of $Ca^{2+}$ regulation of muscle contraction.

[1] Graduate School of Frontier Biosciences, Osaka University, 1-3 Yamadaoka, Suita, Osaka 565-0871, Japan. [2] RIKEN Center for Biosystems Dynamics Research and SPring-8 Center, 1-3 Yamadaoka, Suita, Osaka 565-0871, Japan. [3] JEOL YOKOGUSHI Research Alliance Laboratories, Osaka University, 1-3 Yamadaoka, Suita, Osaka 565-0871, Japan. *email: keiichi@fbs.osaka-u.ac.jp; tfujii@fbs.osaka-u.ac.jp

Muscle contraction occurs through mutual sliding between the thick and thin filaments[1,2] by repeated association and dissociation of myosin head and actin filament coupled with ATP binding, hydrolysis and release by myosin head[3]. In striated muscles, such as skeletal and cardiac muscles, the thick and thin filaments form a regular hexagonal lattice within each sarcomere, which is the contractile unit repeating along the entire muscle cell, and muscle contraction is regulated by intracellular $Ca^{2+}$ concentration[4]. The thin filament consists of actin, tropomyosin (Tm), and troponin (Tn) in 7:1:1 stoichiometry, and Tn is composed of three subunits: troponin C (TnC), the $Ca^{2+}$-binding regulatory subunit; troponin I (TnI), the inhibitory subunit; and troponin T (TnT), the Tm-binding subunit. In the resting state of muscle, the intracellular $Ca^{2+}$ concentration is lower than μM level by a $Ca^{2+}$ ATPase on the sarcoplasmic reticulum (SR) membrane pumping $Ca^{2+}$ into the SR. The regulatory $Ca^{2+}$ is dissociated from TnC, making the thin filament in the inactive $Ca^{2+}$-off state in which the interactions of myosin head to actin filament is prohibited. Upon $Ca^{2+}$ release from the SR, however, $Ca^{2+}$ binding to TnC causes conformational changes of Tn and a shift in the azimuthal position of Tm on actin filament to allow actin–myosin interactions. It is thought that the azimuthal position of Tm controls the access of myosin head to its binding surface on actin filament[5]. At a low $Ca^{2+}$ concentration, Tm is located at a position to block myosin binding on actin filament, and at a high $Ca^{2+}$ concentration, Tm slightly moves its position to uncover the myosin binding site[5]. The binding of $Ca^{2+}$ to TnC triggers this structural switching to the $Ca^{2+}$-on state. This regulatory mechanism is specific to striated muscles.

TnC is the $Ca^{2+}$ binding subunit, TnI inhibits actomyosin ATPase, and TnT has a long N-terminal region binding to Tm. Tm is a long α-helical coiled-coil protein that binds over seven actin subunits on one of the two long-pitch helical strand of actin filament and forms head-to-tail connections with adjacent Tm molecules to produce a continuous helical strand along actin filament. Tn binds to a specific position of Tm, making its periodicity 385 Å corresponding to the actin strand of seven subunits, thus making the stoichiometry of actin, Tm, and Tn 7:1:1[6].

Structure of the thin filament has been analyzed by electron microscopy (EM) and image analysis[5,7–11]. Structures of component proteins have also been revealed either fully or partially[12–15]. However, the structure of the entire thin filament is not yet fully resolved at sufficiently high resolution to give insights into the regulatory mechanism. Structural analysis of the thin filament is difficult because its symmetry and periodicity very much differ from those of actin filament. Actin filament is a helical assembly of actin subunits with a helical pitch of 51.9 Å and an axial repeat of 27.6 Å[15] and can be recognized as a two-start long-pitch helix, but the repeating unit is one actin subunit. However, the repeating unit of the thin filament is a pair of large multimeric complexes, consisting of seven actin subunits, one Tm and one Tn in each of the long-pitch helical strand of actin with their axial stagger of 27.6 Å to each other, and therefore the repeating period is 386.4 Å. Therefore, conventional helical image reconstructions using the helical symmetry of actin filament with a unit repeat of 27.6 Å cannot be applied. The density of Tm survives in helical image analysis simply because the Tm structure has seven homologous repeating units that approximately follows the actin symmetry along its long coiled coil structure[5,7]. It is also difficult to align images of Tn molecules on the thin filament due to their relatively small size. The molecular weight of Tn is approximately 80,000, and this hetero-trimeric complex is not compactly folded, producing only a weak contrast in electron cryomicroscopy (cryoEM) images of the thin filament.

Here we have determined the structure of the entire thin filament from human cardiac muscle in the absence and presence of $Ca^{2+}$ by fully utilizing the recent technical advances in cryoEM and image analysis. These two structures reveal how Tm and Tn bind to actin filament and how $Ca^{2+}$ binding to TnC causes conformational changes of Tn and Tm to allow actin–myosin interactions for muscle contraction.

## Results

**Structural stabilization for cryoEM imaging**. The sample for cryoEM image analysis was prepared by reconstitution with actin filament from skeletal muscle, recombinant cardiac Tm and Tn as a ternary complex of TnC, TnI, and TnT (Supplementary Fig. 1). In negative staining EM images, Tm and Tn are well retained on actin filament (Supplementary Fig. 1d) but were mostly dissociated from actin filament during cryoEM grid preparation by blotting and plunging it into liquid ethane, likely due to partial denaturation at the air–water interface as well as strong fluid flow by blotting. We used recombinant Tm with alanine–serine or alanine–alanine–serine extension to mimic the N-terminal acetylation to stabilize the binding of Tm to actin filament[16], as was successfully used in solving the cryoEM structure of actin filament–Tm complex at high resolution[17]. Addition of a 21-fold excess of Tm and Tn over actin monomer to the sample solution of stoichiometric mixture also improved the efficiency in image collection of intact thin filament by markedly reducing the proportion of bare actin filament in the cryoEM images.

**CryoEM data collection and image analysis**. To visualize the density of Tn in the three-dimensional (3D) image reconstruction, helical image reconstruction conventionally used for the structural analysis of actin filament cannot be used because it smears out the Tn density[5,7]. We therefore extracted image segments of the thin filament from cryoEM images with a box size of 440 Å, which is large enough to cover the entire length of the repeating unit consisting of 14 actin subunits and a pair of Tn and Tm complexes and treated it as an asymmetric single particle. As described in details in Methods and Supplementary Fig. 2, we first selected segment images with Tn around the center by template-based auto-picking with templates images manually selected from 2D class averages. Initial 3D image reconstructions and classifications were carried out with a relatively loose spherical mask large enough to cover a pair of Tn core domains on the two long-pitch actin strands because use of a tight mask dramatically reduced the number of remaining segments and even the images with a pair of Tn densities visibly attached to the filament were discarded. We then made the mask gradually tighter, and for the final 3D classification, completely masked out actin and Tm from our density map to carry out focused classification[18], thereby selecting segment images with a pair of Tn densities both at similar levels (Supplementary Fig. 2).

**3D map and modeling of Tn core domain and Tm**. The structure of the thin filament in the $Ca^{2+}$ free state is shown in Fig. 1 together with a typical cryoEM image of the thin filament with a periodic binding of Tm and Tn. The thin filament is oriented with the pointed end of actin filament top in Fig. 1 and all the other figures. The resolution of the 3D reconstruction is 6.6 Å at a Fourier Shell Correlation of 0.143 (Supplementary Fig. 3). The quality and correctness of the 3D density map can be easily evaluated by nicely fitted atomic model of F-actin solved using the helical symmetry (PDB:3MFP)[15]. This also indicates that the binding of Tm and Tn alters neither the helical symmetry and axial repeat distance of actin filament nor the conformation of actin.

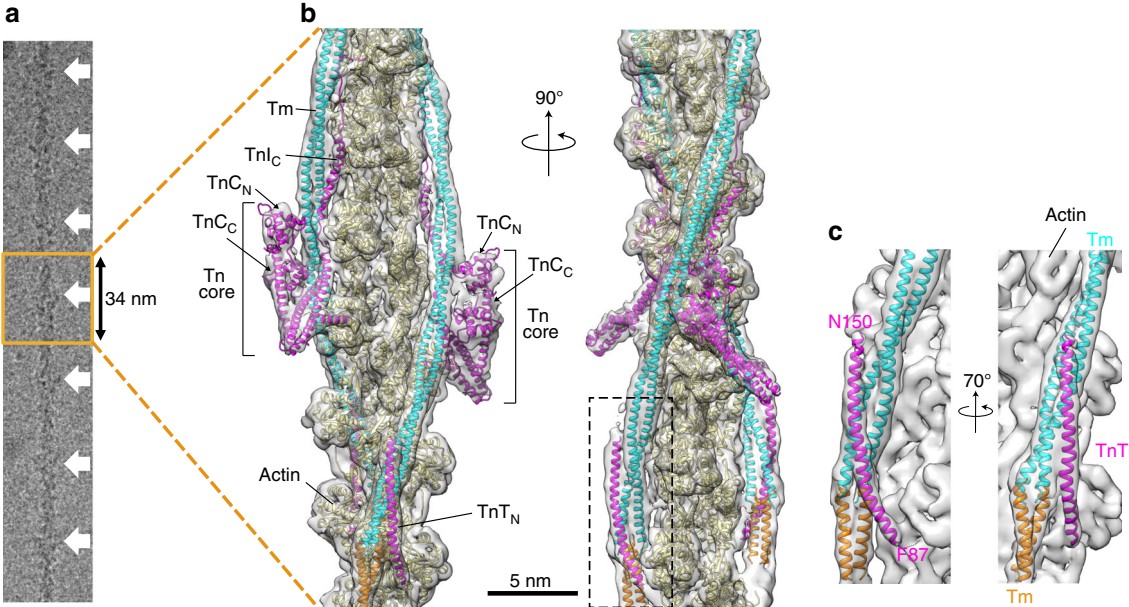

**Fig. 1 Structure of human cardiac thin filament in the Ca$^{2+}$ free state. a** a typical cryoEM image of the thin filament with periodic Tn binding along actin filament. **b** The model of the thin filament fitted into the cryoEM density map in two different views. Approximately 12 actin subunits are shown with a pair of Tm coiled coils and a pair of the Tn ternary complexes consisting of the Tn core, TnI$_C$ extended upward from the core and TnT$_N$ attached closely to the head–tail junction of Tm near the bottom. **c** Magnified images of TnT$_N$ complexed with the head–tail junction of Tm. The models are colored as: actin, beige; Tn, purple; and Tm, light blue and orange. Actin filament is oriented with its pointed end top including all the other figures.

The density of the core domain of Tn together with a long two-stranded coiled coil of Tm is visualized clearly on each strand of actin filament, with one of them at a higher position than the other by the axial rise of actin subunit along actin filament (Fig. 1b). Unfortunately the Tn crystal structures solved to date consists of TnC and fragments of TnI and TnT and covers only 65% of the total Tn chain lengths[13,14], but they could be used as template models to fit into the density map of the Tn core domain. We found one of the crystal structures of human cardiac Tn solved in the Ca$^{2+}$ bound state and deposited in the PDB relatively recently (PDB ID code: 4Y99) showed a better fit into our density map than the others. So we used major parts of this model that cover the following three regions: TnT 199–272 residues forming a short and a long α-helix; TnI 32–137 residues consisting of two long α-helices; and TnC 91–161 residues forming its C-lobe (TnC$_C$). We fitted the entire model as a rigid body and found good fit of individual α-helices into our density map. Since the N-lobe of TnC (TnC$_N$), which is also present in the crystal structure, did not fit into the density at the top of the Tn core as it was in the crystal because the crystal structure was solved in the Ca$^{2+}$-bound state, TnC$_N$ had to be fitted independently using the crystal structure of skeletal muscle TnC$_N$ in the Ca$^{2+}$-free state (PDB: 1YV0)[14], in which the EF hand of TnC$_N$ (helices 2 and 3) is closed with the regulatory switch helix of TnI (residues 149–160) unbound. The Tn core model thus built binds to each strand of actin filament over two actin subunits (Fig. 1b).

The model of full-length Tm including the head-to-tail junction formed by the N- and C-terminal regions of the two adjacent Tm subunits along actin filament has not been determined because all the cryoEM structural analysis have been done with the helical symmetry of actin filament[7,17]. Single particle image analysis has also been used for negatively stained EM images of the thin filament to visualize the Tn–Tm complex on actin filament but the resolution is limited to around 25 Å, which is too low to resolve the Tm junction and even the Tn core structure as well[10,11]. Therefore, we first constructed a homology

model of Tm based on its crystal structure determined at 7 Å resolution (PDB: 1C1G)[12], fitted it to the density map and then refined it by RosettaCM[19]. Although the resolution of our map is not high enough to identify side chains, the structures of two-stranded coiled coil and the head-to-tail junction are clearly resolved to allow relatively reliable modeling. The model of the head-to-tail junction (Fig. 1c) is very similar to that of the solution NMR structure of model peptides (PDB: 2G9J)[20], with both the N- and C-terminal coiled coils spread apart to allow mutual insertion with 90° rotation of their coiled coil planes to each other. A long rod-shaped density closely associated with this Tm junction was identified as part of the N-terminal chain of TnT (TnT$_N$, also called TnT1), and its length was approximately 90 Å, roughly corresponding to 60 residues of α-helix. We therefore built a long α-helix model with residues 87–150 (Fig. 1c, Supplementary Fig. 4) based on the local Tm–TnT$_N$ interactions available in the crystal structure of a much shorter segment of this region[21], in which residues 83–97 of TnT$_N$ including the RRKEEEE motif (residues 107–121 including RKKEEEE in human cardiac TnT) are involved. Interestingly, the N-terminal side of this Tm-bound TnT$_N$ binds to actin, stabilizing the Tm–Tn binding to actin filament, and this explains why many disease-causing mutations are found in this region of TnT[22].

**Highly elongated shape of the entire Tn structure**. After having the model building completed for actin filament, Tm and Tn core as much as possible, we subtracted the model density from the 3D map to identify the densities of the remaining parts of Tn. The densities are separated into two parts, one (red) above and the other (dark blue) below the Tn core as shown in Fig. 2a. We identified the upper density (red) as a C-terminal region of TnI and the lower one (dark blue) as an N-terminal region of TnT, part of which has already been modeled above, based on the continuity of the densities from the Tn core domain. The C-terminal region of TnI comprising the inhibitory region (residues 137–148), regulatory switch (residues 149–160) and mobile region (residues 163–210) is present as a mostly continuous

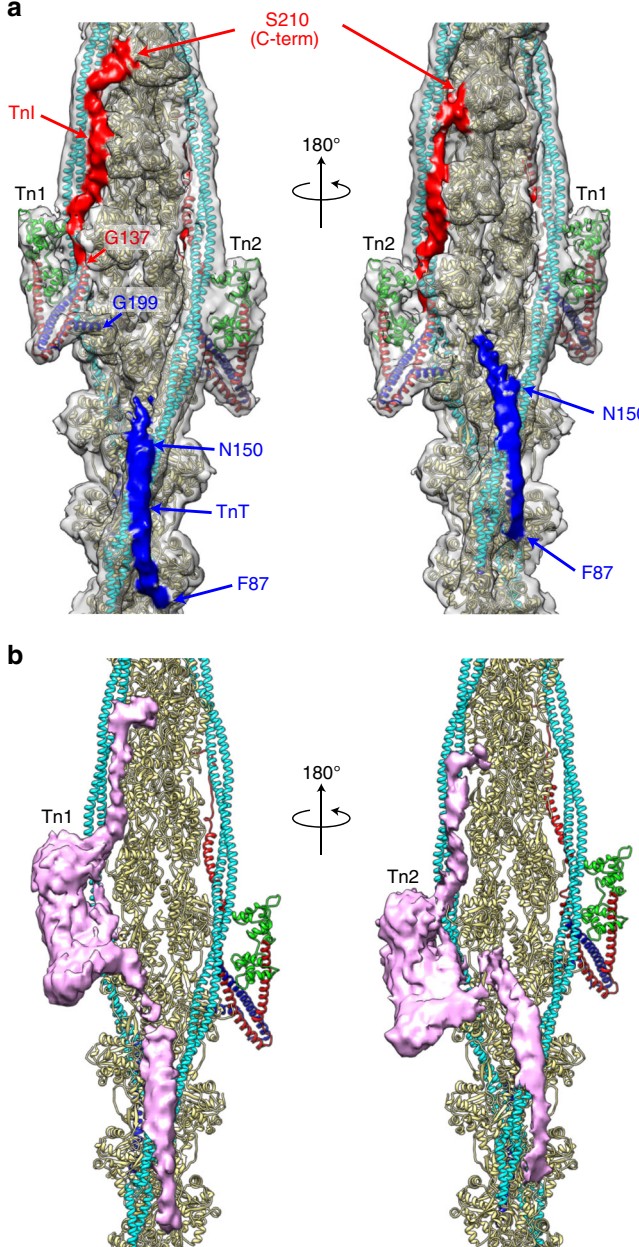

**Fig. 2 The entire structures of the Tn complex with two different conformations. a** The entire structure of the Tn ternary complex is shown for each of the Tn pair, labeled Tn1 and Tn2, in front and back of the thin filament. The difference maps for TnI$_C$ and TnT$_N$ are colored red and dark blue above and below the Tn core, respectively. The difference in the relative positions of the Tn core and TnT$_N$ between the pair is clear while the conformations of the TnI$_C$ chains are the same. **b** The difference map for the entire Tn ternary complex is contoured at a lower level to show the continuous volume of Tn in pink, for each of the Tn pair, again in front and back of the thin filament.

to-tail junction. TnT$_N$ residues 87–150 fitted to the map extends further down beyond the head-to-tail junction and reaches nearly the bottom of the third actin subunits below the Tn core from which the TnT$_N$ density extends. This means that the entire Tn structure is extremely elongated, binding to actin filament over seven actin subunits, of which the upper four are bound by TnI$_C$ and the Tn core and the bottom three of the opposite strand are bound by TnT$_N$ (Fig. 2a). Interestingly, because of this TnT$_N$ density bridging two Tm strands, the distance between the Tn core and the Tm head-to-tail junction is longer on one face of the thin filament (Tn1, the upper one) than the other (Tn2, the lower one) by one actin subunit (Fig. 2a, b). Therefore, the conformations of the TnT$_N$ linker region (residues 151–198) connecting the Tn core and the long N-terminal α-helix formed by TnT$_N$ residues 87–150 are distinct from each other for the pair of the Tn complexes, indicating the flexible nature of this linker region, and the axial extension of Tn2 is shorter than Tn1 by one actin subunit (Fig. 2b). The difference map produced by subtracting the model density of actin filament with Tm from the cryoEM 3D reconstruction is presented at a relatively low contour level to visualize the entire Tn structure of the thin filament in Fig. 2b, clearly showing the highly elongated shape of the Tn complex along actin filament and Tm strand and the conformational difference between Tn1 and Tn2. The sequence regions of Tn covered in the model are indicated in Supplementary Fig. 4.

**Structural comparison of Ca$^{2+}$ free and bound states.** We also carried out cryoEM structural analysis of the thin filament in the Ca$^{2+}$ bound state, obtained a 3D reconstruction at 4.8 Å resolution (Supplementary Fig. 3), built an atomic model and compared it with that of the Ca$^{2+}$ free state (Fig. 3). The overall structures of the entire Tn complexes in a pair are similar to those of the Ca$^{2+}$ free state, showing the same conformational difference between the pair in the TnT$_N$ linker region connecting the Tn core and the long N-terminal α-helix of TnT$_N$, except that the density of the C-terminal region of TnI above the Tn core is missing in the Ca$^{2+}$ bound state (Fig. 2, Supplementary Fig. 5). The model of Tn core in the Ca$^{2+}$ free state fitted quite well into the density map of the Ca$^{2+}$ bound state but TnC$_N$ and the short N-terminal α-helix of TnT needed some modifications. The Tn core in the Ca$^{2+}$ bound state shows a marked change in the orientation by its counter clockwise rotation of about 30° in the horizontal plain with a slight swing up of its distal tip of the IT arm, the major part of Tn core formed by an N-terminal region of TnI (residues 41–136) and a C-terminal region of TnT (residues 199–272) (Supplementary Fig. 4). We fitted a model of TnC$_N$ in the Ca$^{2+}$ bound state[13,14], in which the EF hand of TnC$_N$ is fully open to accommodate the regulatory switch helix of TnI (residues 149–160). Although the conformation of TnT$_N$ below the Tn core, crossing to the opposite actin strand and extending down along and slightly across the Tm coiled coil does not show much changes between the two states, the elongated density of TnI$_C$ above the Tn core, clearly present in the Ca$^{2+}$ free state, extending up along and between actin strand and Tm coiled coil, mostly disappears in the Ca$^{2+}$ bound state (Fig. 3a–d). Even around and within the Tn core, TnC$_N$ and the short N-terminal α-helix of TnT changes its position and orientation relative to the rest of the Tn core (Fig. 3c, d). Most of all, the Tn core shows a large change in its position and orientation together with the azimuthal shift of Tm relative to actin filament, forming more intimate interactions with actin subunits (Fig. 3e). These conformational changes are depicted in Supplementary Movie 1.

As expected from previous observations revealed by EM structural analysis using the helical symmetry of actin filament[5,7,17,23], the position of the long coiled coil of Tm is

density (red) extending up along and between actin strand and Tm coiled coil from the C-terminal end (Gly 137) of the TnI helix within the Tn core. This density, identified as the C-terminal one third of TnI (TnI$_C$), reaches up to the second actin subunit above the one that binds to TnC$_N$. The other one (dark blue) is also highly elongated and is extending back from Gly 199 of TnT across the actin filament strands and down towards Tm on the other actin strand to form a long N-terminal α-helix making a specific interaction with the C-terminal end of Tm near the head-

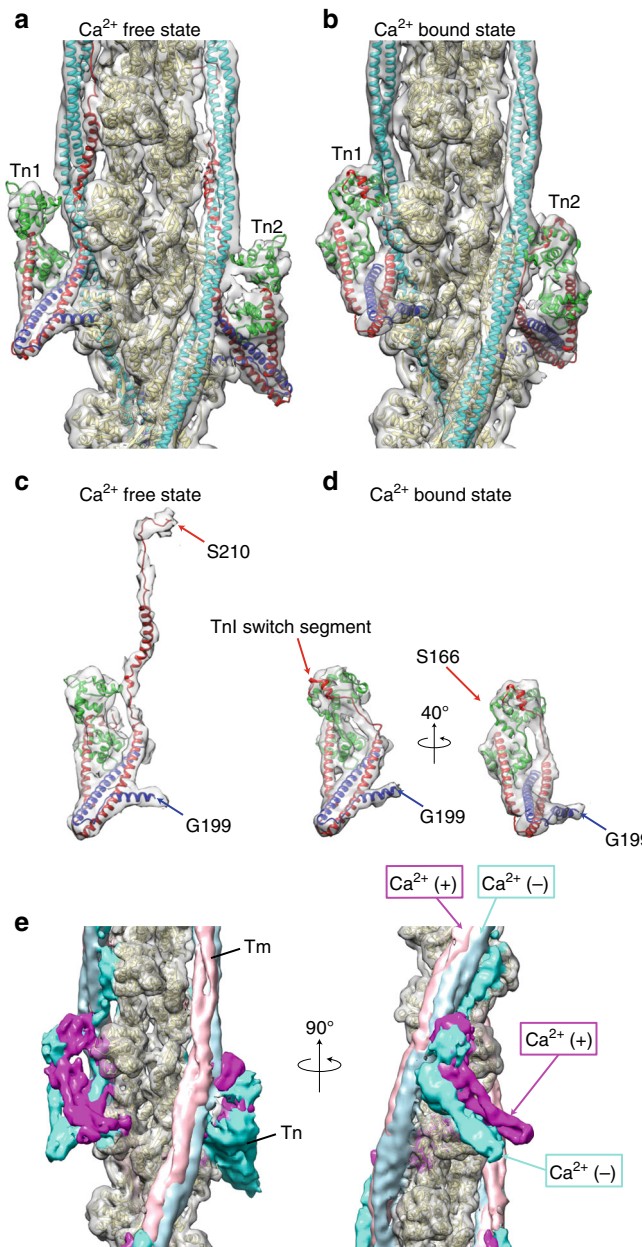

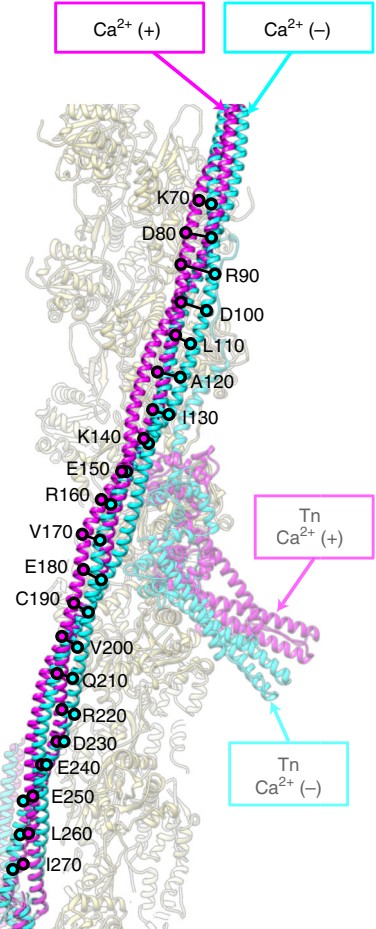

**Fig. 4 Variation in the shift of Tm coiled coil on the surface of actin filament.** Tm coiled coils are colored blue and magenta for the $Ca^{2+}$ free and bound states, respectively. Positions of corresponding $C\alpha$ atoms are indicated by open circles.

**Fig. 3 Structural changes of the thin filament and Tn complex upon $Ca^{2+}$ binding to TnC. a**, **c** $Ca^{2+}$ free state. **b**, **d** $Ca^{2+}$ bound state. **c**, **d** The Tn ternary complex extracted from the thin filament to show the good fit of the model into the difference map. Models are colored as: actin, beige; Tm, light blue; TnI, magenta; TnT, dark blue; and TnC, green. **e** 3D volumes of the thin filament in the two states are superimposed with Tn and Tm in blue and light blue for the $Ca^{2+}$ free state and in magenta and light pink for the $Ca^{2+}$ bound state and actin in beige for both. Two different views are presented.

shifted azimuthally along the surface of actin filament by about 10 Å. But now we clearly see a rolling motion of Tm on the surface of actin filament with their contact points as pivots (Supplementary Movie 1) and also a significant variation in the shift distance along the Tm coiled coil depending on the position along the coiled coil (Fig. 4, Supplementary Fig. 6). The azimuthal shift around the head-to-tail junction is markedly smaller than the other part near and above the Tn core where the shift distance is close to what has been shown in the previous studies by EM helical image analysis of various types of the thin filaments. This

is likely due to $TnT_N$ binding to Tm in this region and its N-terminal side also binding to actin filament, restraining the Tm shift. We then compared the positions of Tm around the Tn core of our thin filament structures in the $Ca^{2+}$ free and bound states with those of Tm in the actin filament–Tm[17] and actomyosin filament–Tm complexes[23] (Fig. 5). While the Tm position in the actin filament–Tm structure without Tn is between those of the $Ca^{2+}$ free and bound states, the one in the actomyosin filament–Tm complexes is further away from the $Ca^{2+}$ free state beyond the $Ca^{2+}$ bound state, indicating that Tn in the $Ca^{2+}$ free state keeps Tm in the position to fully block the access of myosin head to actin filament and, upon $Ca^{2+}$ binding, shifts Tm to a position that allows myosin head access but not to the fully open position to allow the strong binding of myosin head.

## Discussion

The structures of muscle thin filaments analyzed by EM image analysis to date have never been able to visualize the Tn structure in sufficient detail to give us even clues to the $Ca^{2+}$ regulatory mechanism of muscle contraction, either because the resolution is too low to reliably fit the crystal structures of Tn fragments[8–11] or because the use of actin helical symmetry smears out the Tn structure, which does not follow the actin symmetry[5,7,17,23]. In the present study, we applied cryoEM single particle image analysis to human cardiac thin filament in both the $Ca^{2+}$ free and bound states and successfully visualized their structures in

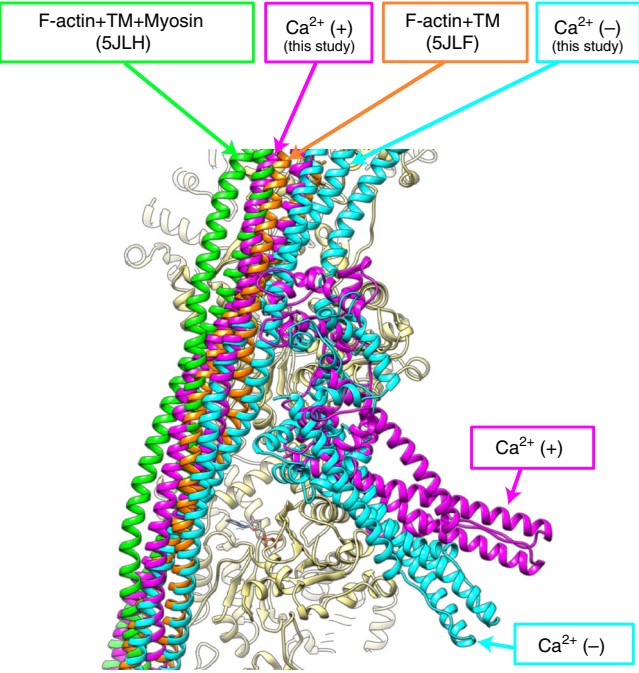

**Fig. 5 Positions of Tm coiled coil in the four different states.** Models are colored as: actin, beige; Tm and Tn in the $Ca^{2+}$ free state, blue; Tm and Tn in the $Ca^{2+}$ bound state, magenta; Tm in the actin filament–Tm structure, orange; and Tm in the actomyosin filament–Tm structure, green. The model 5JLH[23] is a human actomyosin–tropomyosin complex composed of the motor domain of human non-muscular myosin-2C, cytoplasmic γ-actin, and cytoplasmic tropomyosin 3.1, and the model 5JLF[23] is composed of rabbit skeletal muscle actin and recombinant α-tropomyosin from *Mus musculus* with an alanine–serine extension in its N-terminus.

sufficient detail to allow discussion of the regulatory mechanism. The quality of the 3D maps was high enough to allow the reliable docking of available crystal structures of Tn fragments part by part together with Tm and actin filament to build the atomic models of almost entire thin filament in the two functional states. Since the crystal structure of the Tn core (PDB:4Y99)[13] except for the TnC N-lobe (consisting of TnI residues 42–136, TnT residues 203–271, TnC residues 93–161) fitted to the 3D map quite well in both maps, the positions and orientations of the Tn core are now defined much more accurately than ever before. Beside the details of the Tn core structures, the full length Tm structures with the head–tail junction as well as the N-terminal extension of TnT and the C-terminal extension of TnI below and above the Tn core, respectively, are well visualized to allow the modeling of these parts albeit the accuracy is somewhat limited. With these two models of the thin filament in the $Ca^{2+}$ free and bound states, we can now discuss the $Ca^{2+}$ regulatory mechanism of muscle contraction in detail.

The inhibitory role of the $TnI_C$ chain downstream from Gly 137 has been characterized well by biochemical studies[24–26]. It is now visualized in the structures of the thin filament in the two states. In the $Ca^{2+}$ free state, the EF hand of $TnC_N$ is closed[14], and the regulatory switch of TnI (residues 149–160) released from its binding site in the $Ca^{2+}$ bound open EF hand of $TnC_N$ is tightly bound to actin and Tm (Figs. 3a, 6a), and this also leads to the tripartite binding of the remaining chain of $TnI_C$ beyond this switch segment to actin filament and Tm above the Tn core (Figs. 2, 3a), stabilizing the Tm coiled coil in the OFF position ("blocked"[27–29]) that blocks the myosin head access to actin filament (Fig. 5). This $TnI_C$ chain as well as the Tn core themselves also strongly contribute to the blocking of myosin head

access over four myosin binding sites of actin filament (Fig. 2). Upon $Ca^{2+}$ binding to $TnC_N$, its EF hand can be opened to accommodate the regulatory switch helix of TnI[30] (Figs. 3b, 6b), depleting the regulatory switch and eventually the entire $TnI_C$ chain from actin filament and Tm to shift Tm to the ON position ("closed"[27–29]), which is slightly further away from the position than that of the actin filament–Tm complex (Fig. 5). This means that Tn in the $Ca^{2+}$ bound state does not simply release Tm from its restrained position in the $Ca^{2+}$ free state to a Tn free position but more actively determines the Tm position to make the myosin binding site of actin filament more readily accessible for myosin head. $TnC_N$ gets much closer to actin in the $Ca^{2+}$ bound state, coming into the position between actin and Tm occupied by the regulatory switch of TnI in the $Ca^{2+}$ free state (Fig. 6b), probably pushing Tm further away to make myosin binding site more open. Then, myosin head binding to actin filament causes further azimuthal shift of Tm to the fully ON position ("open") as has been discussed in many different studies[27–29]. A schematic diagram of the $Ca^{2+}$ regulatory mechanism is presented in Fig. 7. Although the $Ca^{2+}$ regulatory mechanism of muscle contraction is thus revealed as the structural changes of Tn and Tm on actin filament upon $Ca^{2+}$ depletion and binding, much higher resolution structures of the thin filament would be needed for its confirmation as well as for more quantitative examination of the regulatory mechanism, for which further development of cryoEM methods would be required.

## Methods

**Protein expression and purification.** G-actin from rabbit skeletal muscle (Muscle Actin > 95 % pure, #AKL95, cytoskeleton, Inc.) was polymerized in a polymerization buffer (10 mM Tris-HCl pH 7.5, 100 mM KCl, 2 mM $MgCl_2$, 0.2 mM ATP, 1 mM DTT) at room temperature for 2 h to prepare actin filament. Human cardiac α-tropomyosin with an N-terminal alanine–serine extension was expressed in *E. coli* strain KRX (Promega). *E. coli* cells were harvested by low speed centrifugation (11,800$g$), resuspended and sonicated in a lysis buffer (20 mM Tris-HCl pH 7.5, 100 mM NaCl, 2 mM EGTA, 5 mM $MgCl_2$). The *E. coli* proteins and cell debris were precipitated by heating the sample solution to 80 °C for 15 min and centrifugation at 146,000$g$. The pH of the supernatant was decreased to 4.6 by adding 0.1 M HCl with stirring for 20 min to precipitate Tm. The solution was centrifuged at 48,000$g$, and the pellet was resuspended in a running buffer for ion exchange chromatography (20 mM Tris-HCl pH 7.0, 100 mM NaCl) with 10 mg $mL^{-1}$ DNase, incubated for 2 h and then centrifuged at 108,000$g$ to remove aggregates. The supernatant was applied to ion exchange chromatography using a HiTrap Q column (GE healthcare), eluted with a 100 mM–1 M NaCl gradient[16]. The three component proteins of human cardiac troponin, TnC, TnI, and TnT, were expressed and assembled together in *E. coli* KRX strain to form the ternary complex according to the protocol by Lassalle[31] as follows. TnC was cloned into pET21b(+) with an N-terminal Strep-tag II and a Tobacco Etch Virus (TEV) protease cleavage site using SLiCE method[32]. TnI and TnT were cloned into the pACYCDuet vector. His8-tag-superfolder green fluorescent protein (sfGFP) and TEV protease cleavage site were fused to TnI at the N-terminus, and His6-tag and TEV protease cleavage site were attached to TnT at the N-terminus. These two vectors were co-transformed into *E.coli* strain KRX (Supplementary Fig. 1a). TnC, TnI and TnT were co-expressed, and the ternary complex formed in the cell was purified by NiNTA affinity chromatography in an elution buffer containing 50 mM Tris-HCl (pH 8.0), 300 mM NaCl, a 100–400 mM imidazole gradient, and by subsequent Strep-tag II Tactin affinity chromatography in an elution buffer containing 150 mM NaCl, 100 mM Tris–HCl (pH 8.0), 1 mM EDTA, 2 mM D-biotin. Strep-tag II, His6-tag and His8-tag-sfGFP were digested by TEV protease, and the digested products were applied to a NiNTA affinity column to collect the Tn ternary complex in the flow through. The Tn ternary complex was concentrated and purified by size exclusion chromatography on a Superdex 200 increase 10/300 GL columns (GE healthcare) equilibrated with a SEC buffer (20 mM Tris-HCl pH7.5, 300 mM NaCl, 1 mM EDTA, 2 mM DTT). All the primer sequences used in this study are listed in Supplementary Table 2.

**Assembly of the thin filament and cryoEM grid preparation.** Actin filament, Tm and the Tn ternary complex, prepared as in the previous section, were mixed at protein concentrations of 7.5 μM, 21 μM and 21 μM, respectively, and the solution was incubated for 10 min at room temperature to prepare the thin filament. The molar ratio of the mixture is 1:3:3 for actin, Tm and Tn, which is a large excess in the amounts of Tm and Tn to actin, since the stoichiometry of actin, Tm and Tn in the thin filament is 7:1:1. This is to ensure the complete decoration of actin filament by Tm and Tn. The thin filament solution was diluted to a concentration of

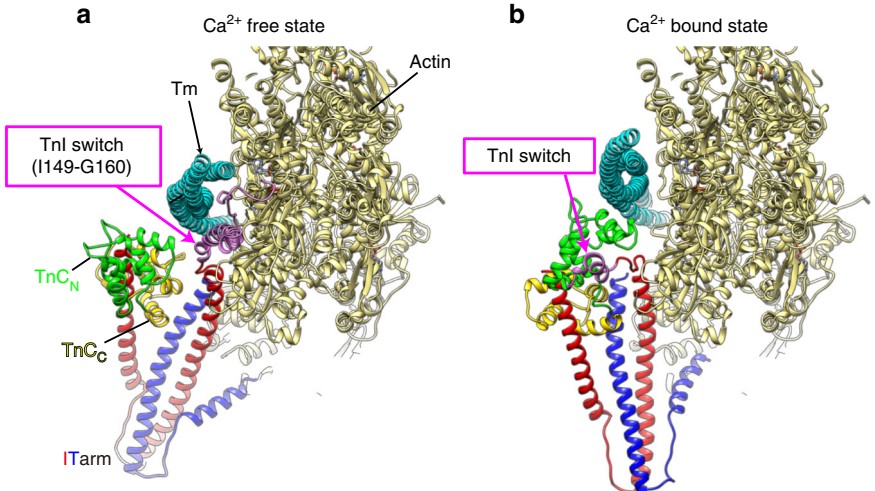

**Fig. 6 Structural changes of the thin filament and Tn complex upon Ca$^{2+}$ binding to TnC. a** Ca$^{2+}$ free state, **b** Ca$^{2+}$ bound state. Model colors are the same as in Fig. 3 except for TnC$_C$ colored brown. The models are viewed obliquely from the pointed end of actin filament.

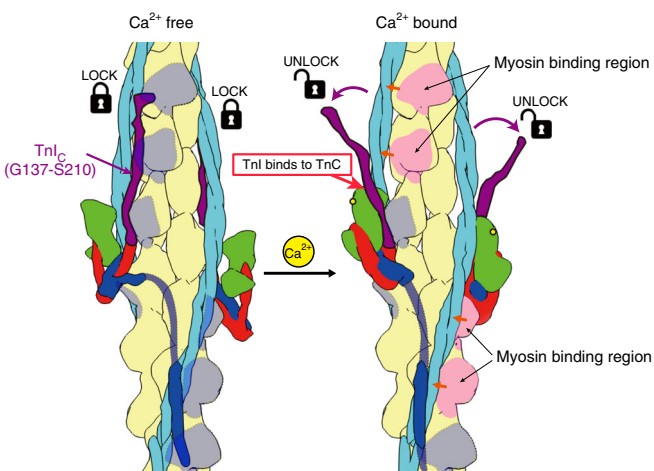

**Fig. 7 Schematic diagram of the Ca$^{2+}$ regulatory mechanism of muscle contraction.** The structural changes between Ca$^{2+}$ free and bound states are shown. In the Ca$^{2+}$ free state, TnI$_C$ (dark purple) binds to actin (beige) and Tm (light blue) above the Tn core (green, red and blue), covering the myosin head binding sites. But entire TnI$_C$ dissociates from them upon Ca$^{2+}$ binding to TnC$_N$ by the binding of a short N-terminal portion of TnI$_C$ to TnC$_N$. This causes Tm to move around on the actin filament surface (short orange arrows) together with TnT$_N$ near the head-to-tail junction of Tm, thereby exposing some of the myosin head binding sites shaded in light gray to allow actin–myosin interactions (light pink).

0.03 mg mL$^{-1}$ in a buffer containing 10 mM Tris-HCl pH 7.5, 100 mM KCl, 2 mM MgCl$_2$, either with 3 mM EGTA for Ca$^{2+}$ free state or 3 mM CaCl$_2$ for Ca$^{2+}$ free state. A 3.0 μl sample solution was applied to a glow discharged holey carbon molybdenum grid (Quantifoil R1.2/1.3), incubated for 10 s and blotted for 7 s from the backside with a filter paper (Whatman No.1) at 4 °C with 80% humidity, before vitrification by plunging the grid into liquid ethane using a vitrification devise, EM-GP (Leica).

**Electron cryomicroscopy**. CryoEM images were recorded under a defocus range of 0.7–1.8 μm on a prototype of CRYO ARM 200 electron cryomicroscope (JEOL Ltd.), equipped with an Ω-type in-column energy filter and a Schottky-type field emission gun operated at 200 kV, using a K2 Summit direct electron detector camera (Gatan Inc.) with the counting mode at a nominal magnification of ×50 K, corresponding to a pixel size of 1.1 Å on the specimen. Each image was fractionated into 52 frames of every 0.3 s, at a dose rate of ~1.2 e$^-$ pixel$^{-1}$ second$^{-1}$, to

accumulate a total dose of ~65 e$^-$ Å$^{-2}$. Dose fractionated cryoEM images were recorded using an automated data acquisition program, JADAS (JEOL Ltd.).

**Image processing**. In total 13,272 and 2,880 images were collected for the samples of Ca$^{2+}$ free and Ca$^{2+}$ bound states, respectively. Motion correction was carried out by MotionCor2[33], defocus values were measured by CTFFIND 4.1[34], and subsequent image analysis was carried out using RELION 3.0[35]. We first manually picked a few thousands of segment images of the thin filament with a box size of 400 pixels, binned the image to a box size of 100 pixels (4.4 Å pixel$^{-1}$), carried out 2D classification, and selected 10 class averages as templates for subsequent template-based auto-picking. Then, template-based auto-picking of segment images with the same box size and a box distance of 27.6 Å was performed. A total of 6,405,681 and 1,527,182 segment images were then extracted for Ca$^{2+}$ free and Ca$^{2+}$ bound states, respectively. In the early stage of image processing, we carried out 2D classification and 3D classification at a pixel size of 4.4 Å to speed up the calculations. We discarded bad segment images by particle sorting and 2D classification. First 3D reconstruction and 3D auto-refine was done with the helical symmetry of actin filament to align the filament axes. Then, 3D classification with 14 classes was carried out without helical symmetry, with a spherical mask of a diameter of 170 Å to include a pair of Tn core domain within the mask, to obtain the initial structure of the thin filament. We selected classes in which the pair of Tn core domain was positioned near the center of segment image and performed 3D reconstruction. Using the obtained 3D map, we made a relatively loose envelop of the thin filament and extracted a central 45% part of its axial length to use as a mask for the subsequent 3D classification (mask 1 in Supplementary Fig. 5). The 3D classification was performed with 14 classes and then made a tighter envelop as a mask (mask 2 in Supplementary Fig. 5) for the next 3D classification with a smaller number of classes less than 10. We selected classes in which the pair of Tn core domain was positioned at the center of segment image, by the numbers of segment images remained were 251,696 and 86,343 for Ca$^{2+}$ free and Ca$^{2+}$ bound states, respectively. These images were re-extracted with a box size of 200 pixels at a pixel size of 2.2 Å, and these segment images were aligned. After the alignment, we masked out actin filament and Tm (mask 3 in Supplementary Fig. 5), focused only on the pair of troponin and carried out 3D classification for more strict selection of segment images. The segment images were classified by 3D classification, first into 8 classes, second into 5 and finally into 3, and the best class in the final round contained 21,588 and 23,374 for Ca$^{2+}$ free and Ca$^{2+}$ bound states segment images, which were subjected to 3D auto-refine and Post-process to reconstruct the final 3D map. The details of the image processing procedure are described in Supplementary Fig. 2. Summary of cryoEM data collection and image analysis is presented in Supplementary Table 1.

**Modeling of actin filament, Tm, Tn core, TnI and TnT**. First, a model of actin filament we built on our 3.6 Å resolution map (PDB ID: 5JLF)[23] was fitted into the 3D map of the thin filament as a rigid body. For modeling Tm structure, a homology model of Tm was constructed with the full length Tm crystal structure determined at 7 Å resolution (PDB: 1C1G)[12] as a template and fitted it into the 3D map. The Tn core domain was fitted by one of the crystal structures of human cardiac Tn in the Ca$^{2+}$ bound state (PDB: 4Y99)[13]. Since the resolution of the density of TnC$_N$ was not high enough to perform flexible fitting, we divided the Tn core into three domains, TnC$_N$, TnC$_C$, and IT arm, and treated them as rigid bodies. Since the Tm-Tm junction was clearly resolved in the 3D map, it was

possible to place the N- and C-termini of four TM chains into the junction to build a model of Tm for its entire length. We then subtracted the model densities of actin filament and Tm from the 3D map of the thin filament to produce a difference map, which revealed the densities for the remaining chains of an N-terminal region of TnT ($TnT_N$) and the C-terminal region of TnI ($TnI_C$). For modeling $TnT_N$, an α-helix model of $TnT_N$ residues 87–150 was built by MODELLER[36], and the interactions between the C-terminal end of rabbit skeletal Tm and a short fragment of chicken skeletal TnT revealed in the crystal structure (PDB: 2Z5H)[21] was used to build a model of Tm–$TnT_N$ complex. For $TnI_C$, the difference map showed an elongated density along actin filament and Tm coiled coil above the Tn core only in the $Ca^{2+}$ free state. We used the long C-terminal α-helix visualized in one of the crystal structure of human cardiac Tn (PDB: 1J1E)[13] to fit into the difference map and modeled the remaining C-terminal region as an extended chain. We used RosettaCM[19] for all the modeling and refinement to remove clashes between actin, Tm and Tn and keeping their stereochemistry and used UCSF Chimera for the preparation of all the figures[37].

**Reporting summary.** Further information on research design is available in the Nature Research Reporting Summary linked to this article.

## Data availability

The cryoEM volumes have been deposited in the Electron Microscopy Data Bank under accession codes EMD-0728 and EMDB-0729, and the atomic coordinates have been deposited in the Protein Data Bank under accession codes 6KN7 and 6KN8, for the structures of the thin filament in the $Ca^{2+}$ free and bound states, respectively. Other data are available from the corresponding authors upon reasonable request.

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

## Acknowledgements

The authors thank Takayuki Kato for technical help in automated cryoEM data collection by JADAS software with CRYO ARM 200 electron microscope. This work was supported JST PRESTO Grant number JPMJPR12L6 to T.F and by JSPS KAKENHI Grant number 25711010 to T.F and 25000013 to K.N and was also supported by Platform Project for Supporting Drug Discovery and Life Science Research (BINDS) from AMED under Grant Number JP19am0101117 to K.N., by the Cyclic Innovation for Clinical Empowerment (CiCLE) Grant Number JP17pc0101020 from AMED to K.N. and by JEOL YOKOGUSHI Research Alliance Laboratories of Osaka University to K.N.

## Author contributions

T.F. and K.N. initiated the project, T.F. designed the protein expression plasmid and purification protocol, Y.Y. and T.F. performed biochemical experiments, cryoEM imaging and analysis and model building. Y.Y., T.F. and K.N. carefully studied the models and wrote up the manuscript together.

## Competing interests

The authors declare no competing interests.
