## [Peer Review File · Nature Communications]

Reviewers' comments:

Reviewer #1 (Remarks to the Author):

This is a very solid paper that provides a significant advance in understanding thin filament regulation. The authors have used asymmetric reconstructions of cryoEM images of thin filaments in both the high and low Ca states. This surmounts the problems of previous work which either was at extremely low resolution in negative stain or required the imposition of actin helical symmetry in ice. While it is likely that higher resolution studies will take place in the future, this paper is worthy of publication with only minor changes.

p. 4)"probably due to partial denaturation at the air-water interface". This is likely, but in addition there are strong fluid flow and compression forces that will be experienced by such filaments, as discussed and shown in Galkin et al., Structure (2015).

Fig. 4) It should be indicated in the legend or the figure that Tn is also shown.

Extended Data Fig. 3) The high Ca FSC does not go to 0.0 for the masked volume as it does in the low Ca state. The curve is also truncated at $\sim 1/(4.5 \text{ \AA})$, while the Nyquist frequency is close to $1/(2 \text{ \AA})$.

Extended Data Fig. 4) Showing the C-C distances to one thousandth of an \AA appears silly when the atomic models have been built into maps at $\sim 5\text{-}6 \text{ \AA}$ resolution. It would seem much more reasonable to not even show a tenth of an \AA given the uncertainties arising from the limited resolution.

Reviewer #2 (Remarks to the Author):

Review of the manuscript entitled “Cardiac muscle thin filament structures reveal calcium regulatory mechanism” by Yurika Yamada, Keiichi Namba and Takashi Fujii

The authors report the structures of human cardiac muscle thin filament in the absence and presence of Ca^{2+} . Their work reveals the structure of a C-terminal region of troponin I and that of an N-terminal region of troponin T in complex with the head-to-tail junction of tropomyosin together with the troponin core on actin filament. In the presence of Ca^{2+} the position of tropomyosin is shifted azimuthally along the surface of actin filament by about 10 Å. This shift is for the first time observed to be associated with a rolling motion of tropomyosin on the surface of actin filament with the overlap regions acting as pivot points. Moreover, the authors report that the shift distance along the tropomyosin co-filament is not uniform but is greatly reduced in the proximity of the overlap regions.

The work is original and highly relevant. I complement the authors on their scientific and technical achievements, which are of the highest quality. The work will have a marked influence on the field.

I greatly enjoyed reading the manuscript and I had no problems understanding the text. Nonetheless, the manuscript would certainly benefit from another round of careful language editing. The following comments are aimed to help the authors in improving their manuscript.

Page 1

Muscle contraction occurs through mutual sliding between the thick and thin filaments by repeated association and dissociation of myosin head and actin filament coupled with ATP binding, hydrolysis and release by myosin head¹.

This description of the mechanism of muscle contraction omits the power-stroke that occurs while the myosin motor is strongly bound to F-actin. Moreover, the authors of reference 1 postulated for the first time the tight coupling between the chemical and mechanical actomyosin reaction cycle but they did not reveal the mechanism.

Upon Ca^{2+} release from the SR, however, Ca^{2+} binding to TnC causes conformational changes of Tn and Tm on actin filament to allow actin-myosin interactions.

Tropomyosin undergoes a troponin-mediated change in position but not in conformation upon Ca^{2+} release from the SR.

Page 3

Structural analysis of the thin filament is difficult because its symmetry and periodicity very much differ from those of actin filament, which is a helical assembly of actin subunits with a helical pitch of 51.9 Å and an axial repeat of 27.6 Å¹³.

This statement is confusing in its present form. I would like to suggest to change it in the following way.

The actin filament is a two-start long-pitch helix. Both in F-actin and in the thin filament the spacing between actin monomers in a single helix of an actin filament corresponds to a distance of 55 Å. As

the two actin chains are staggered by half a subunit period with respect to one another, the rise per monomer of actin along each chain corresponds to 27.5 Å. In contrast, the repeating unit of the thin filament corresponds to one complete helical repeat of the actin filament, a 385 Å segment of the two-start long-pitch helix comprising per strand seven actin monomers and one Tm-Tn complex.

Page 4

The sample of human cardiac muscle thin filament was prepared by reconstitution with actin filament from skeletal muscle, recombinant cardiac Tm and Tn as a ternary complex of TnC, TnI and TnT (Extended Data Fig. 1).

As the authors used rabbit skeletal muscle and recombinant tropomyosin constructs with alanine-serine or alanine-alanine-serine extensions, they should not claim that they reconstituted a human cardiac muscle thin filament.

Addition of excessive amounts of Tm and Tn to the sample solution of stoichiometric mixture also helped efficient image collection of intact thin filament by reducing bare actin filament in the cryoEM images.

Replace with:

Addition of a 21-fold excess of Tm and Tn over actin monomers improved image collection of intact thin filament by markedly reducing the proportion of bare actin filament in the cryoEM images.

Page 5

We fitted the entire model as a rigid body “body”

Page 8

The Tn core in the Ca²⁺ bound state shows a marked change in the orientation by its counter clockwise rotation of about 30° in the horizontal plain with a slight swing up of its distal tip of the IT arm formed by the N-terminal ends of TnI-TnT coiled coil.

Explain “IT arm”

Page 14

Human cardiac α -tropomyosin (Tm) with an N-terminal alanine-serine extension was expressed in *E. coli* and purified based on the method previously reported²¹.

Reference 21 is not the correct reference for the expression and purification of α -tropomyosin.

For modeling TnT_N, an α -helix model of TnT_N residues 90 – 151 was built by MODELLER³⁴, and the interactions between the C-terminal end of rabbit skeletal Tm and a short fragment of chicken skeletal TnT revealed in the crystal structure¹⁹ was used to build a model of Tm- TnT_N complex.

Missing PDB code => 2Z5H

Figure 5:

The authors should provide more detailed information about the compared structures, what is shown and what is omitted. Thus, 5JHL should be clearly identified as human actomyosin–tropomyosin (ATM) complex, composed of the motor domain of human non-muscular myosin-2C (NM-2C), cytoplasmic gamma-F-actin and cytoplasmic tropomyosin 3.1 and 5JLF composed of rabbit skeletal muscle actin and recombinant alpha-tropomyosin from Mus musculus with an alanine-serine extension.

Our responses to the comments by the reviewers are listed below.

We also made minor modifications by adding a few sentences in p. 8, a schematic diagram explaining the Ca^{2+} regulatory mechanism as Figure 7 and Supplementary Figure 4, and by changing the order of Supplementary Figures 5 and 6.

To Referee #1:

This is a very solid paper that provides a significant advance in understanding thin filament regulation. The authors have used asymmetric reconstructions of cryoEM images of thin filaments in both the high and low Ca states. This surmounts the problems of previous work which either was at extremely low resolution in negative stain or required the imposition of actin helical symmetry in ice. While it is likely that higher resolution studies will take place in the future, this paper is worthy of publication with only minor changes.

Thank you very much for the favorable comment on this study.

p. 4)"probably due to partial denaturation at the air-water interface". This is likely, but in addition there are strong fluid flow and compression forces that will be experienced by such filaments, as discussed and shown in Galkin et al., Structure (2015).

We modified the text as "likely due to partial denaturation at the air-water interface as well as strong fluid flow by blotting".

Fig. 4) It should be indicated in the legend or the figure that Tn is also shown.

We added labels for Tn in the figure.

Extended Data Fig. 3) The high Ca FSC does not go to 0.0 for the masked volume as it does in the low Ca state. The curve is also truncated at $\sim 1/(4.5 \text{ \AA})$, while the Nyquist frequency is close to $1/(2 \text{ \AA})$.

The FSC curve is not truncated and is actually shown to the Nyquist frequency. In the final stage of image analysis including classification, refinement and reconstruction, images were binned to $2.2 \text{ \AA}/\text{pixel}$, and therefore the Nyquist frequency is $1/4.4 \text{ \AA}$, which is 0.23 \AA^{-1} . We added a more detailed description of the pixel size in the Methods section to make it clear.

Extended Data Fig. 4) Showing the C-C distances to one thousandth of an \AA appears silly when the atomic models have been built into maps at $\sim 5\text{-}6 \text{ \AA}$ resolution. It would seem much more reasonable to not even show a tenth of an \AA given the uncertainties arising from the limited resolution.

We agree and rounded off the distances to a tenth of an \AA .

Reviewer #2 (Remarks to the Author):

The authors report the structures of human cardiac muscle thin filament in the absence and presence of Ca^{2+} . Their work reveals the structure of a C-terminal region of troponin I and that of an N-terminal region of troponin T in complex with the head-to-tail junction of tropomyosin together with the troponin core on actin filament. In the presence of Ca^{2+} the position of tropomyosin is shifted azimuthally along the surface of actin filament by about 10 \AA . This shift is for the first time observed to be associated with a rolling motion of tropomyosin on the surface of actin filament with the overlap regions acting as pivot points. Moreover, the authors report that the shift distance along the tropomyosin co-filament is not uniform but is greatly reduced in the proximity of the overlap regions.

The work is original and highly relevant. I complement the authors on their scientific and technical achievements, which are of the highest quality. The work will have a marked influence on the field.

I greatly enjoyed reading the manuscript and I had no problems understanding the text. Nonetheless, the manuscript would certainly benefit from another round of careful language editing. The following comments are aimed to help the authors in improving their manuscript.

Thank you very much for the precise and appropriate recognition of and comments on the value of this work.

Page 1

Muscle contraction occurs through mutual sliding between the thick and thin filaments by repeated association and dissociation of myosin head and actin filament coupled with ATP binding, hydrolysis and release by myosin head¹.

This description of the mechanism of muscle contraction omits the power-stroke that occurs while the myosin motor is strongly bound to F-actin. Moreover, the authors of reference 1 postulated for the first time the tight coupling between the chemical and mechanical actomyosin reaction cycle but they did not reveal the mechanism.

We did not mention the power-stroke mechanism of muscle contraction because it is not the subject of this study. We added two references (Refs. 1 and 2) for the mutual sliding between the thick and thin filaments in the beginning of the sentence.

Upon Ca²⁺ release from the SR, however, Ca²⁺ binding to TnC causes conformational changes of Tn and Tm on actin filament to allow actin-myosin interactions.

Tropomyosin undergoes a troponin-mediated change in position but not in conformation upon Ca²⁺ release from the SR.

We changed the last half of the sentence as follows.

“conformational changes of Tn and a shift in the azimuthal position of Tm on actin filament to allow actin-myosin interactions.”

Page 3

Structural analysis of the thin filament is difficult because its symmetry and periodicity very much differ from those of actin filament, which is a helical assembly of actin subunits with a helical pitch of 51.9 Å and an axial repeat of 27.6 Å¹³.

This statement is confusing in its present form. I would like to suggest to change it in the following way.

The actin filament is a two-start long-pitch helix. Both in F-actin and in the thin filament the spacing between actin monomers in a single helix of an actin filament corresponds to a distance of 55 Å. As the two actin chains are staggered by half a subunit period with respect to one another, the rise per monomer of actin along each chain corresponds to 27.5 Å. In contrast, the repeating unit of the thin filament corresponds to one complete helical repeat of the actin filament, a 385 Å segment of the two-start long-pitch helix comprising per strand seven actin monomers and one Tm-Tn complex.

We changed this part as follows.

“Actin filament is a helical assembly of actin subunits with a helical pitch of 51.9 Å and an axial repeat of 27.6 Å¹⁵ and can be recognized as a two-start long-pitch helix, but the repeating unit is one actin subunit. However, the repeating unit of the thin filament is a pair of large multimeric complexes, consisting of seven actin subunits, one Tm and one Tn in each of the long-pitch helical strand of actin with their axial stagger of 27.6 Å to each other, and therefore the repeating period is 386.4 Å. Therefore, conventional helical image reconstructions using the helical symmetry of actin filament with a unit repeat of 27.6 Å cannot be applied.”

Page 4

The sample of human cardiac muscle thin filament was prepared by reconstitution with actin filament

from skeletal muscle, recombinant cardiac Tm and Tn as a ternary complex of TnC, TnI and TnT (Extended Data Fig. 1).

As the authors used rabbit skeletal muscle and recombinant tropomyosin constructs with alanine-serine or alanine-alanine-serine extensions, they should not claim that they reconstituted a human cardiac muscle thin filament.

We deleted the phrase “human cardiac muscle thin filament” and modified the sentence as “The sample for cryoEM image analysis was prepared by reconstruction ...”. But since the amino acid sequence of rabbit skeletal muscle actin has 99% homology with human cardiac actin, the structure we analyzed is nearly identical to that of the human cardiac muscle thin filament.

Addition of excessive amounts of Tm and Tn to the sample solution of stoichiometric mixture also helped efficient image collection of intact thin filament by reducing bare actin filament in the cryoEM images.

Replace with:

Addition of a 21-fold excess of Tm and Tn over actin monomers improved image collection of intact thin filament by markedly reducing the proportion of bare actin filament in the cryoEM images.

We modified the sentence accordingly.

We fitted the entire model as a rigid body..... “body”

Corrected.

The Tn core in the Ca²⁺ bound state shows a marked change in the orientation by its counter clockwise rotation of about 30° in the horizontal plain with a slight swing up of its distal tip of the IT arm formed by the N-terminal ends of TnI-TnT coiled coil.

Explain “IT arm”

We modified the sentence as follows.

“the IT arm, the major part of Tn core formed by an N-terminal regions of TnI (residues 41 – 136) and a C-terminal region of TnT (residues 199 – 272) (Supplementary Fig. 4).”

Page 14

Human cardiac α -tropomyosin (Tm) with an N-terminal alanine-serine extension was expressed in *E. coli* and purified based on the method previously reported²¹.

Reference 21 is not the correct reference for the expression and purification of α -tropomyosin.

We replaced it with the correct one (Ref. 16).

Monteiro P. B., Lataro R. C., Ferro J. A., Reinach Fde C. Functional alpha-tropomyosin produced in *Escherichia coli*. A dipeptide extension can substitute the amino-terminal acetyl group. *J. Biol. Chem.* **269**, 10461-10466 (1994).

Page 17

For modeling TnTN, an α -helix model of TnTN residues 90 – 151 was built by MODELLER³⁴, and the interactions between the C-terminal end of rabbit skeletal Tm and a short fragment of chicken skeletal TnT revealed in the crystal structure¹⁹ was used to build a model of Tm- TnTN complex.

Missing PDB code => 2Z5H

We inserted the PDB code in the sentence.

Figure 5:

The authors should provide more detailed information about the compared structures, what is shown and what is omitted. Thus, 5JHL should be clearly identified as human actomyosin–tropomyosin (ATM) complex, composed of the motor domain of human non-muscular myosin-2C (NM-2C), cytoplasmic gamma-F-actin and cytoplasmic tropomyosin 3.1 and 5JLF composed of rabbit skeletal muscle actin and recombinant alpha-tropomyosin from *Mus musculus* with an alanine-serine extension.

We added a few more labels in the figure and the following sentence in the legend.

“The model 5JLH²³ is a human actomyosin–tropomyosin complex composed of the motor domain of human non-muscular myosin-2C, cytoplasmic γ -actin and cytoplasmic tropomyosin 3.1, and the model 5JLF²³ is composed of rabbit skeletal muscle actin and recombinant α -tropomyosin from *Mus musculus* with an alanine-serine extension in its N-terminus.”